# Proteomics-driven noninvasive screening of circulating serum protein panels for the early diagnosis of hepatocellular carcinoma

Xiaohua Xing [1,4], Linsheng Cai[1,2,4], Jiahe Ouyang[1], Fei Wang[1], Zongman Li[1], Mingxin Liu[1], Yingchao Wang[1], Yang Zhou[1], En Hu[1], Changli Huang[2], Liming Wu [1,3] ✉, Jingfeng Liu [2] ✉ & Xiaolong Liu [1] ✉

Early diagnosis of hepatocellular carcinoma (HCC) lacks highly sensitive and specific protein biomarkers. Here, we describe a staged mass spectrometry (MS)-based discovery-verification-validation proteomics workflow to explore serum proteomic biomarkers for HCC early diagnosis in 1002 individuals. Machine learning model determined as P4 panel (HABP2, CD163, AFP and PIVKA-II) clearly distinguish HCC from liver cirrhosis (LC, AUC 0.979, sensitivity 0.925, specificity 0.915) and healthy individuals (HC, AUC 0.992, sensitivity 0.975, specificity 1.000) in an independent validation cohort, outperforming existing clinical prediction strategies. Furthermore, the P4 panel can accurately predict LC to HCC conversion (AUC 0.890, sensitivity 0.909, specificity 0.877) with predicting HCC at a median of 11.4 months prior to imaging in prospective external validation cohorts (No.: Keshen 2018_005_02 and NCT03588442). These results suggest that proteomics-driven serum biomarker discovery provides a valuable reference for the liquid biopsy, and has great potential to improve early diagnosis of HCC.

Hepatocellular carcinoma (HCC) ranks fourth in cancer mortality worldwide, and chronic cirrhosis caused by hepatitis virus (mainly Hepatitis B and C Virus) and metabolic diseases (mainly alcoholic liver disease and diabetes) is the major risk factor for HCC[1,2]. Although surgery remains an effective therapy for HCC patients according to the HCC treatment guidelines, most patients are diagnosed at advanced clinical stage due to the lack of early symptoms and thus suffer from poor outcomes[3]. Thus, early screening and diagnosis of HCC still remain a clinical dilemma.

Current strategies for HCC diagnosis include imaging (CT/MRI), serum protein biomarkers (alpha-fetoprotein (AFP), protein induced by vitamin K absence or antagonist-II (PIVKA-II, namely Des-gamma-carboxy prothrombin) and histopathology, which are difficult to accurately diagnose early-stage HCC due to empirical limitations, restricted sensitivity or invasive detection modalities[4,5]. Serum and plasma are routinely collected in patients with liver symptoms and reflect changes in liver function, making them ideal for liquid biopsy with great safety, simplicity and suitable for large populations with long-period follow-up[6,7], and many circulating liquid biopsy tumor biomarkers such as ctDNA[8], cfDNA[9,10], metabolites[11], and proteins[12–14] are developed rapidly. Plasma or serum proteins, as the ultimate bearers and effectors of human biological activities, are the common study objects in biomarker development. The FDA has approved over 100 plasma or serum proteins and some serum protein biomarkers have been tested for long-term clinical applications[15,16]. Therefore, system-wide discovery of serum protein biomarkers for early diagnosis

[1]The United Innovation of Mengchao Hepatobiliary Technology Key Laboratory of Fujian Province, Mengchao Hepatobiliary Hospital of Fujian Medical University, Fuzhou 350025, China. [2]Department of Hepatopancreatobiliary Surgery, Fujian Cancer Hospital, Clinical Oncology School of Fujian Medical University, Fuzhou 350000, China. [3]Department of Hepatobiliary and Pancreatic Surgery, The First Affiliated Hospital, School of Medicine, Zhejiang University, Hangzhou 310003, China. [4]These authors contributed equally: Xiaohua Xing, Linsheng Cai. ✉e-mail: wlm@zju.edu.cn; drjingfeng@fjmu.edu.cn; xiaoloong.liu@gmail.com

of HCC would be very attractive, and these global data could be used to build up machine learning-based classification models for the early diagnosis of HCC.

Mass spectrometry (MS)-based proteomics is in principle an ideal tool for biomarker discovery. However, proteomic analysis of serum or plasma has been challenging because of low protein concentrations and a wide dynamic range of protein abundances, resulting in low quantification precision, throughput, and limited proteome depth[16–18]. Recent advances in MS-based proteomics have greatly improved the depth and breadth of serum and plasma proteins, and extended its impact in biomedical and clinical studies[19]. Data independent acquisition based MS (DIA-MS) could effectively avoid the masking effect of high abundance proteins (HAP) on low abundance proteins (LAP), and improve detection efficiency and sample reproducibility; therefore, it has been widely used in the development of tumor serum biomarkers[20]. Furthermore, circulating proteomic panels for diagnosis and risk stratification of various tumors were developed using the targeted proteomic strategy, avoiding the restrictions of antibodies[21,22]. The specificity for the identification and quantification of hundreds and even thousands of proteins in serum or plasma samples makes it suitable in principle for the identification and validation of biomarkers. Many research groups have performed a series of biologically meaningful proteomic studies in clinical samples from various clinical cohorts using the DIA + PRM workflow[23–26]. For HCC biomarker discovery, the DIA + PRM strategy has only been applied to very small clinical cohorts, and there is a lack of screening and validation studies in large clinical cohorts.

In this study, using serum as a liquid biopsy, we performed a staged MS-based discovery-verification-validation proteomics workflow in 1002 individuals to screen HCC diagnosis biomarkers, from which a biomarker panel was developed by learning machine for diagnosis of HCC patients. Furthermore, the clinical significance of this panel for early diagnosis and even early predicting of HCC was further evaluated in a prospective cohort. The aim of this research was to reveal the change of serum proteins in HCC patients, discovering valuable serum protein biomarkers for early diagnosis of HCC, and providing valuable data resource for HCC study.

## Results

### Study design and clinical characteristics of serum specimens

To systemically identify and validate potential noninvasive protein biomarkers for HCC diagnosis in serum, we performed a staged MS-based discovery-verification-validation proteomics workflow for this study (Fig. 1). For discovery cohort, 320 individuals including HCC (n = 163), liver cirrhosis (LC, n = 53), basic liver diseases (BLD, n = 64, including 16 chronic hepatitis B (CHB), 18 alcoholic liver disease (ALD) and 30 non-alcoholic fatty liver disease (NAFLD) samples) and chronic asymptomatic hepatitis B virus carrier (AsC, n = 40) patients were included for DIA-MS quantitative proteomic analysis. The detailed clinical information is shown in Supplementary Data 1. There were no statistically significant differences in routine indicators such as gender among patients in different groups. However, the indicators reflecting the severity of the liver function decompensation of patients differed significantly among four groups, which was consistent with the progression of the disease from benign to malignant (Supplementary Data 2). The validation cohort included an independent retrospective validation cohort (n = 429, consisting of 210 HCC patients, 115 LC patients, and 104 healthy controls (HC)) and an independent prospective validation cohort (consisting of 253 LC patients, of whom 36 developed HCC during follow-up). The candidate biomarkers were quantified using targeted proteomics based on parallel reaction monitoring (PRM). Furthermore, machine learning models based on early diagnosis panels for HCC were developed and used for the prediction of HCC risk.

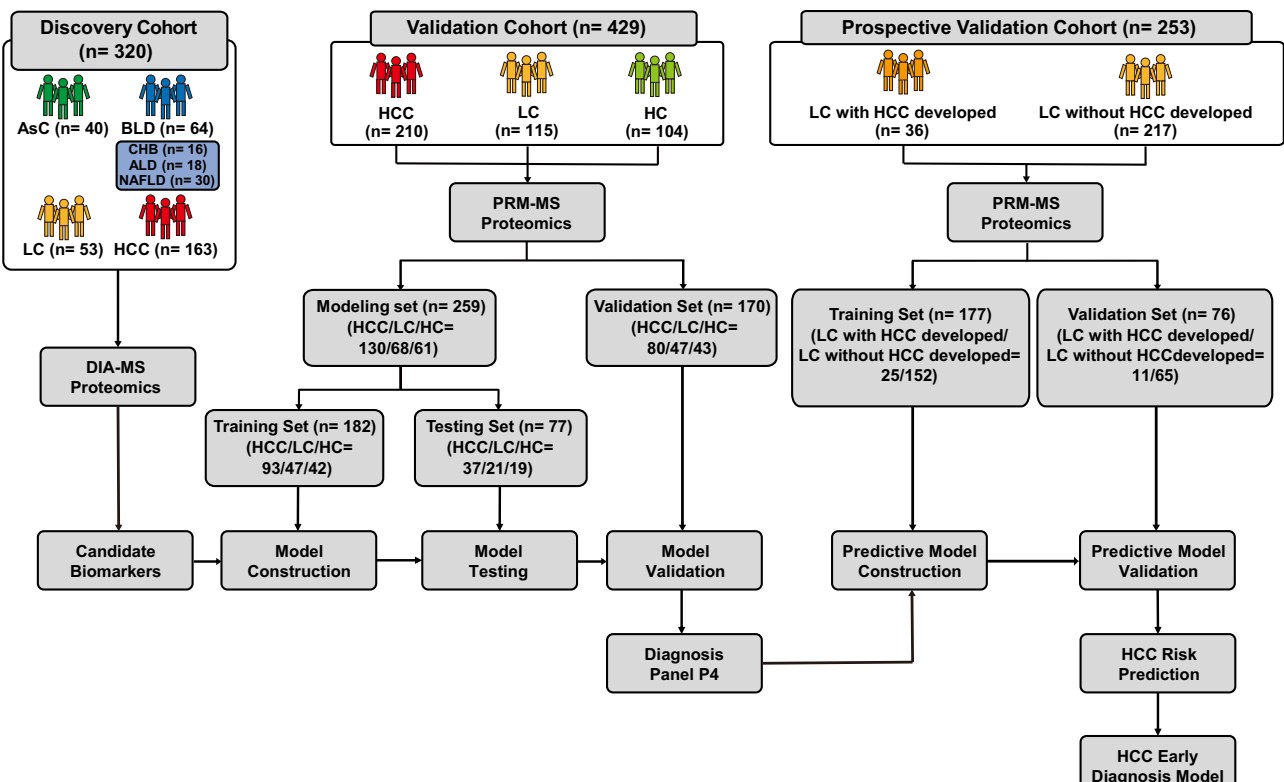

**Fig. 1 | Overall experimental design for biomarker model development.** Large-scale DIA-based proteomics was used to select HCC-related biomarker candidates, which were then validated in an independent validation cohort using PRM-based targeted proteomic approach. HCC diagnosis models were constructed based on machine learning and the efficacy of the models for HCC risk prediction was assessed through prospective long-term follow-up of LC patients.

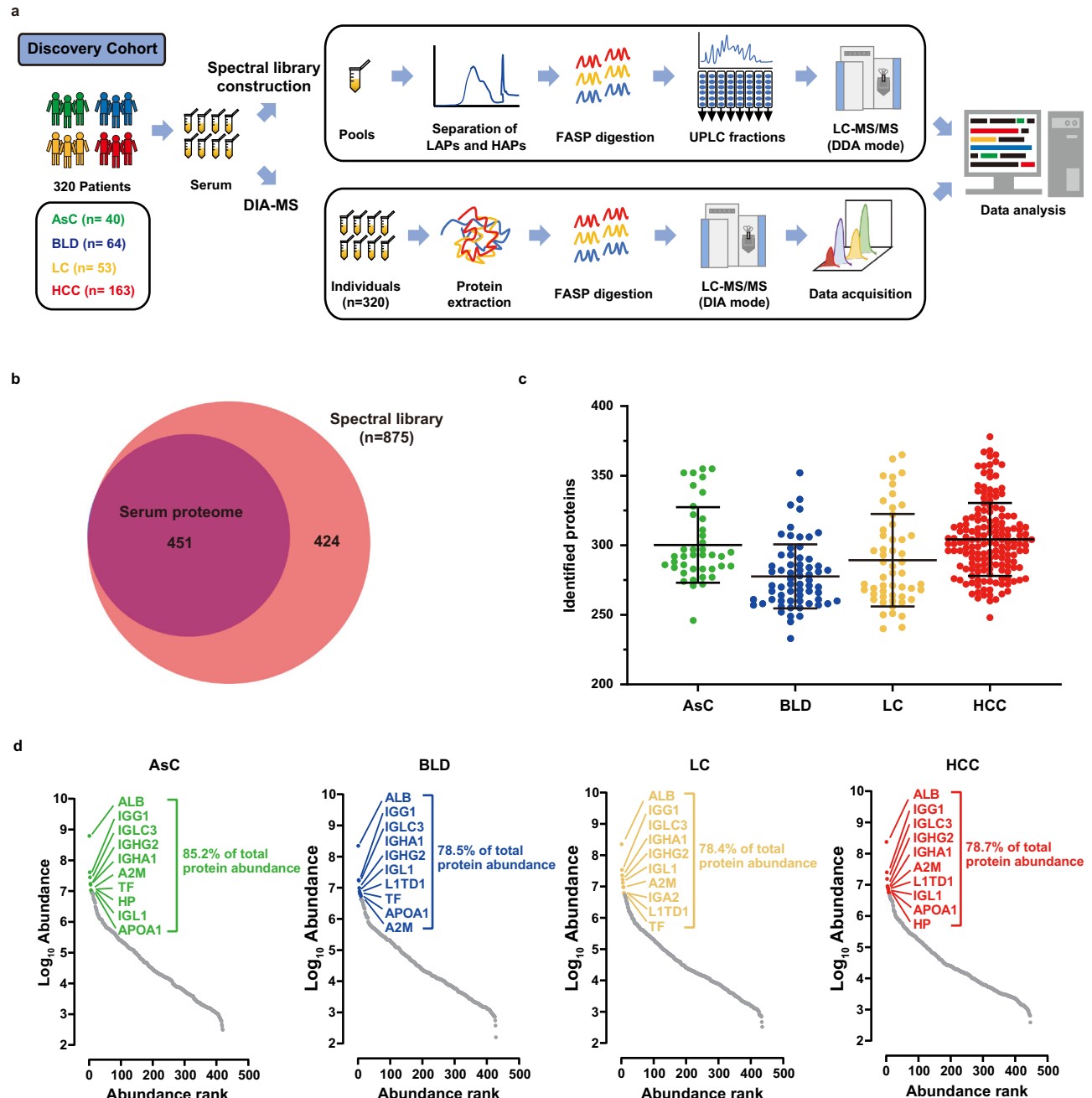

**Fig. 2 | MS-based serum proteomic analysis of discovery cohort. a** Overview of serum proteomics by DIA-MS. **b** Comparison of the number of proteins identified in the serum proteome and the spectral library. **c** Number of proteins identified and quantified with a 1% FDR in four groups (AsC, *n* = 40 biologically independent samples; BLD, *n* = 64 biologically independent samples; LC, *n* = 53 biologically independent samples; HCC, *n* = 163 biologically independent samples). Data represent mean ± SD. **d** Proteins identified in the 4 groups were ranked according to their median intensity. The top ten most abundant proteins are labeled, and their relative contribution to the total protein intensity is indicated. Source data are provided as a Source Data file.

## Proteomic characterization of serum samples

We performed proteome profiling of serum samples using high-throughput DIA-MS-based proteomics strategy (Fig. 2a). To maximize proteome depth and coverage, we generated a hybrid spectral library consisting of 128 fractions of pooled serum samples from DDA and 320 individual serum samples from DIA. The hybrid spectral library contained 875 proteins, of which 82 proteins were detected only in the DIA-MS data (Supplementary Fig. 1a). The majority of the library proteins (85.9%; 752/875) were reported in the Plasma Proteome Database (http://plasmaproteomedatabase.org/) (Supplementary Fig. 1b). Using this spectral library, the DIA-MS

analysis acquired 451 quantifiable proteins, which occupied more than half of proteins in the library (51.5%; 451/875) (Fig. 2b, Supplementary Data 3). On average, we quantified 300 (AsC), 278 (BLD), 289 (LC), and 304 (HCC) proteins in each group per serum sample in a single run (Fig. 2c). Our DIA workflow resulted in a comparable serum proteome coverage with previous studies that applied a similar single-run strategy without deleting serum HAPs[27]. The quantification of serum protein intensity spanned over six orders of magnitude, and the top ten most abundant proteins account for about 80% of the serum proteome signal, illustrating the challenge of analysis (Fig. 2d).

## Assessment of the mass spectrometry platform and proteomic data

To assess the quality control of the mass spectrometry platform in our study, we investigated the variables of our workflow by repeatedly measuring a Hela cell protein digest standard sample throughout the process, including DDA-MS and DIA-MS. The quality-control analysis of different replications showed high technical reproducibility for DDA-MS and DIA-MS, with an average number of quantified proteins of 3591 and 4120 (Fig. 3a), coefficients of variations (CVs) of 0.31 and 0.18 (Fig. 3b), and median correlation coefficients of 0.92 and 0.92 in DDA-MS and DIA-MS, respectively (Supplementary Fig. 1c, d). These results demonstrated the consistent stability of the mass spectrometry platform.

331 proteins in standards were also found in the serum samples, with correlation coefficients ranging from 0.96–0.99 (median: 0.99) (Supplementary Fig. 1e), which were comparable to the standards. To further assess the quantitative accuracy of the serum proteome experiments, we also performed technical replicates analysis of six serum samples (HCC, $n = 2$; LC, $n = 2$; CHB, $n = 2$) in the middle and at the end of the project. Notably, repeat experiments with the same samples have good reproducibility, with a low CV (range: 0.06–0.22; median: 0.11) and a high level of correlation (range: 0.94–0.99; median: 0.95) (Fig. 3c, Supplementary Fig. 1f), thus demonstrating the consistent stability of the serum proteomic experiments.

Furthermore, the CVs values of the four groups were significantly higher than those in the technical replicates (Fig. 3d), which combined with the correlation of identified proteins among four groups revealed the high heterogeneity within patients, especially within the LC group and HCC group (Supplementary Fig. 1g). Consistent with clinical perceptions, AFP and PIVKA-II were higher in HCC than that in non-HCC patients (Supplementary Fig. 1h), and MS quantitative proteomic results of AFP showed a high correlation with clinical antibody-based assays (Fig. 3e, f). When we used Yoden index threshold to classify MS-based AFP quantitation into positive and negative, 82.2% of the patients were consistent with the results that defined by clinical AFP or PIVKA-II antibody assay (Fig. 3g). These results strongly affirmed the high quality of our proteomic data.

## Differentially abundant proteins and functional alterations related to HCC

To further screen meaningful diagnostic biomarkers for HCC, 201 immunoglobulins were excluded from further analyses. A total of 17 up-regulated and 17 down-regulated proteins differed in HCC/AsC, HCC/BLD, and HCC/LC comparisons were used for further analysis (Supplementary Fig. 2a, b, Supplementary Data 4). The expression profiles of these proteins clearly showed intergroup differences and trended with disease severity, with the most dramatic differences in the HCC group (Fig. 4a, b).

As expected, most proteins located in extracellular space, extracellular exosome, extracellular region, and blood microparticle, which was consistent with the characteristics of serum proteins. These dys-regulated proteins mainly enriched in the biological process of immunity and inflammation, as well as in molecular functions associated with activation of multiple receptors and various enzymatic activities related to tumorigenesis and development. Moreover, the enriched pathways like complement and coagulation cascades, NOD-like receptor signaling pathway, NF-kappa B signaling pathway, Toll-like receptor signaling pathway, TNF signaling pathway, and leukocyte transendothelial migration, indicating that HCC was likely to promote its own development by regulating a variety of receptors or pathways related to immune and inflammatory (Fig. 4c).

In addition, PPI network analysis revealed three highly connected clusters involving cell proliferation and apoptosis (blue), cell adhesion and recognition (red), complement activation and innate immunity (green) (Fig. 4d). In the blue and green clusters, the abundance of up-regulated proteins changed more dramatically, suggesting that these proteins might played a dominant role in HCC proliferation, development, and migration. In the green cluster, the abundance of most proteins decreased, suggesting that HCC might have some inhibitory regulation of the immune system. Candidate proteins for further validation were mainly selected from these three clusters.

## Verification of serum candidate biomarkers using PRM-based targeted MS

Based on the HCC-related proteomic and functional alteration revealed in the discovery study, we then sought to develop protein biomarkers that reflect the HCC occurrence with high accuracy. Firstly, the Learning Vector Quantization (LVQ) model was used to evaluate the diagnostic performance of 34 HCC-related differentially abundant proteins, by comparing the accuracy of each protein in identifying HCC patients, 15 proteins with accuracy higher than 0.8 were selected (Fig. 5a). Secondly, the candidate proteins required unique peptides and a good peptide profile matched in DIA-MS data. Finally, 11 candidate biomarkers with unique peptides and aberrant abundance in HCC were proposed for further targeted proteomics analysis (Fig. 5b, Supplementary Table 1). In order to verify the authenticity of candidate biomarker, we further validated the abundance of matched peptides in an independent validation cohort containing 130 HCC patients, 68 LC patients, and 61 HC individuals by PRM-MS. And five peptides could be quantified in more than three pairs of ions matched in the light and heavy labels, and their quantification was statistically significant in the HCC compared with LC and HC groups ($p < 0.05$), which was consistent with the trend of DIA-MS results (Fig. 5c, Supplementary Fig. 3a, b, Supplementary Data 5). Thus, these five proteins were used in different combinations to construct HCC diagnostic models.

## Machine learning-based classification of HCC

To screen the best panel for HCC diagnosis, 130 HCC, 68 LC and 61 HC samples with PRM quantitative data for candidate serum proteins were used to construct a random forest predictive model and to correct the cut-off. Another 80 HCC, 47 LC, and 43 HC samples from the validation set were then introduced to assess the reliability of the model externally. We compared the area under the ROC curve (AUC) for five potential biomarkers and different combinations of permutations in the validation set (Supplementary Table 2). This process resulted in a panel of HABP2 and CD163 with high performance for distinguishing HCC from LC (AUC: 0.935, sensitivity:0.838, specificity: 0.872) and from HC (AUC: 0.977, sensitivity:0.875, specificity: 0.907) (Supplementary Fig. 4a). Notably, it still maintains effective diagnosis in HCC patients who were negative for AFP (<20 ng/mL), PIVKA-II (<40 mAU/mL) and even negative for both AFP and PIVKA-II (Supplementary Fig. 4b-d). Furthermore, it possessed significantly higher score in HCC than that in LC and HC ($p < 0.0001$) and a high diagnostic accuracy of more than 0.85 (Supplementary Fig. 4e-f).

Due to the complementarity of HABP2 + CD163 with AFP and PIVKA-II, we further constructed a 4 protein-based (P4) panel for HCC diagnosis. The P4 panel had significantly improved the performance in diagnosing HCC patients than AFP, PIVKA-II and their combination, with the highest AUC, sensitivity and specificity (Validation Set: HCC/LC: AUC: 0.979, sensitivity: 0.925, specificity: 0.915; HCC/HC: AUC: 0.992, sensitivity: 0.975, specificity: 1.000) (Fig. 6a, Supplementary Fig. 5a, b). Furthermore, P4 panel performed significantly better than PIVKA-II in AFP-negative HCC (Validation Set: HCC/LC: AUC: 0.963 *vs*. 0.893, sensitivity: 0.857 vs. 0.833; HCC/HC: AUC: 0.984 vs. 0.938, sensitivity: 0.952 vs. 0.833) (Fig. 6B, Supplementary Fig. 5c, d), as well as significantly better than AFP in PIVKA-II negative HCC (Validation Set: HCC/LC: AUC: 0.946 vs. 0.868, sensitivity: 0.813 vs. 0.563; HCC/HC: AUC: 0.958 vs. 0.902, sensitivity: 0.875 vs. 0.563) (Fig. 6c,

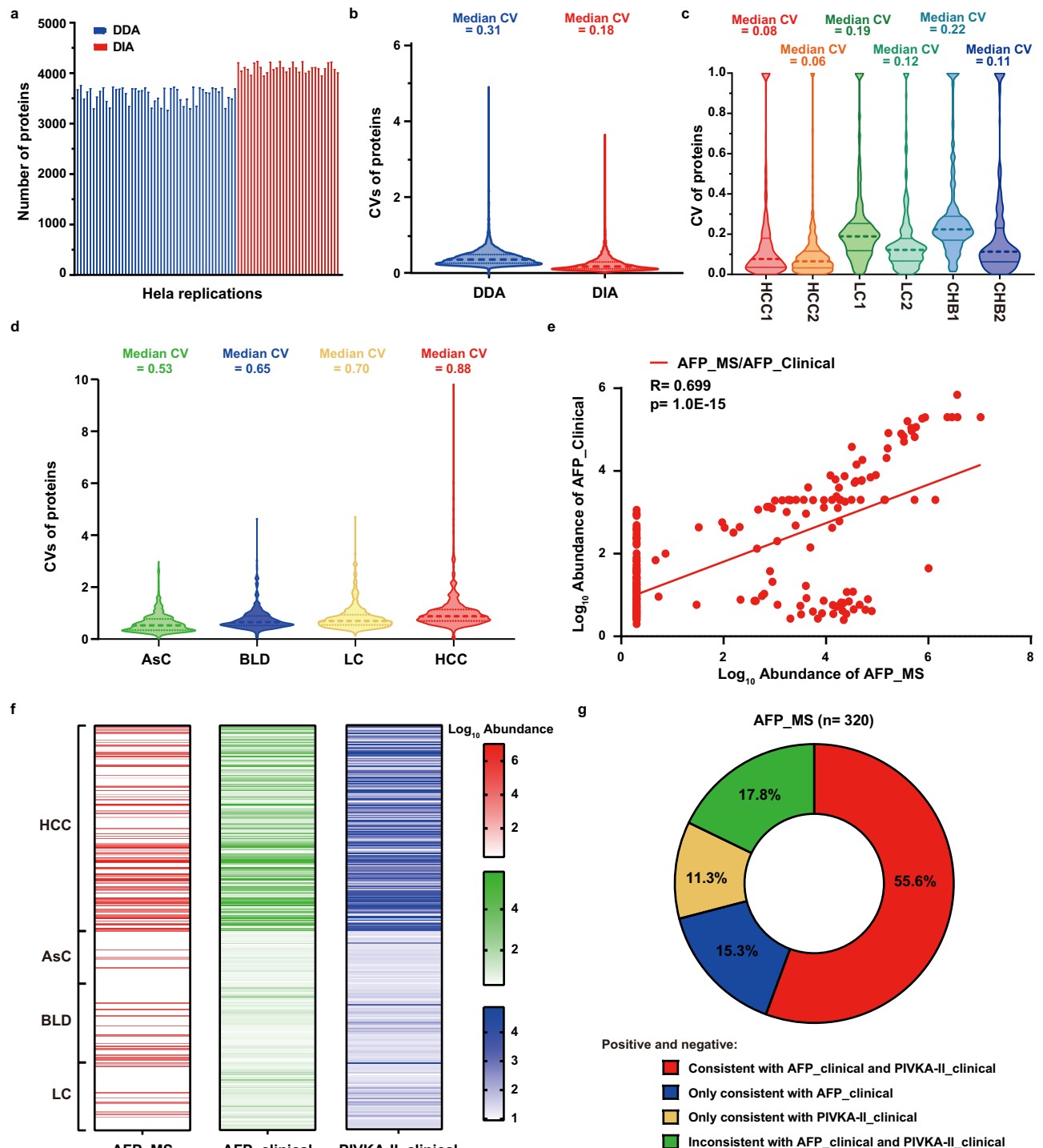

**Fig. 3 | Quality assessment of MS platform and serum proteomics data. a** The protein number of standards through library process and targeted process including data from DDA mode (*n* = 50 independent experiments) and DIA mode (*n* = 32 independent experiments). **b** Distribution of CVs of standards in DDA-MS (*n* = 50 independent experiments) and DIA-MS (*n* = 32 independent experiments). Data represent median, 25% quartile and 75% quartile. **c** Distribution of CVs of technical replicates of 6 serum samples (HCC, *n* = 2 independent experiments; LC, *n* = 2 independent experiments; CHB, *n* = 2 independent experiments) in the middle and at the end of the project. Data represent median, 25%, and 75% quartile. **d** Distribution of CVs of serum samples in four groups (AsC, *n* = 40 biologically independent samples; BLD, *n* = 64 biologically independent samples; LC, *n* = 53

biologically independent samples; HCC, *n* = 163 biologically independent samples). Data represent median, 25%, and 75% quartile. **e** Correlation analysis of AFP quantification results through DIS-MS strategy and clinical serological assays. Pearson's correlation coefficients and *p* value are shown. Significance of linear correlation was determined by one-sided joint hypotheses test. **f** Distribution of AFP_MS, AFP_clinical and PIVKA-II_clinical abundance in four groups. The quantitation data were Log$_{10}$ scaled. **g** Comparison of the consensus rates of AFP_MS negative and positive with AFP_clinical (20 ng/μL) and PIVKA-II_clinical (40 mAU/mL). Threshold of AFP_MS was determined by maximum Youden index. Source data are provided as a Source Data file.

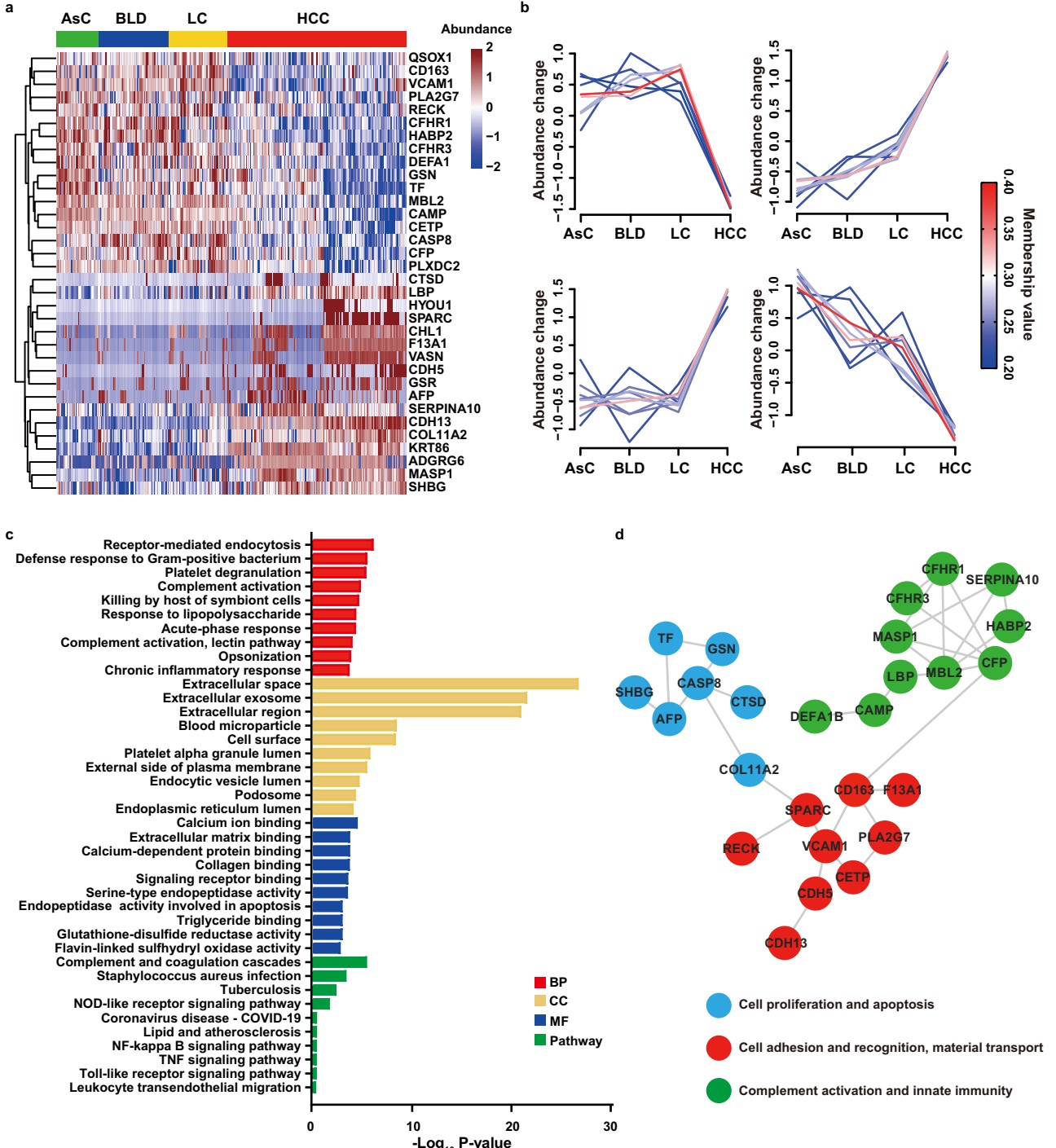

**Fig. 4 | Differentially abundant proteins and functional alterations related to HCC. a** Heatmap representation of abundance profile of differentially abundance proteins in four groups. The quantitation data were Log₂ scaled. **b** Clustering analysis of differentially abundant proteins in four groups using Mfuzz method. The individual line colors reflect the correlation between trends of protein abundance in different groups and median trends in clustered subgroups, with colors closer to red indicating a higher positive correlation, and closer to blue indicating a higher negative correlation. **c** Biological processes (BP), cellular components (CC), molecular functions (MF) and pathways related to the HCC-associated differentially abundant proteins. Top ten terms were shown. Significance of GO items and proteins was determined by hypergeometric test with Benjamini-Hochberg multiple test adjustment. **d** Protein-protein interaction (PPI) network analysis of differentially abundant proteins. Source data are provided as a Source Data file.

Supplementary Fig. 5e, f). Even in AFP negative and PIVKA-II negative HCC patients, the performance of P4 panel remained effective, with AUC of 0.878 (HCC/LC) and 0.904 (HCC/HC) in the validation set (Fig. 6d, Supplementary Fig. 5g, h). In addition, the P4 panel possessed significantly higher score in HCC than that in LC and in HC ($p < 0.0001$) (Fig. 6e, Supplementary Fig. 5i, j), and the high diagnostic accuracy of

HCC patients (HCC/LC: 74/80, 92.1%; HCC/HC: 78/80, 98.4%) indicated the high clinical value of the P4 panel for HCC (Fig. 6f, Supplementary Fig. 5k, l).

Next, we determined the performance of P4 panel in the diagnosis of HCC at the early stage, due to the low sensitivity of existing diagnostic strategies. As the best-performing serum biomarker

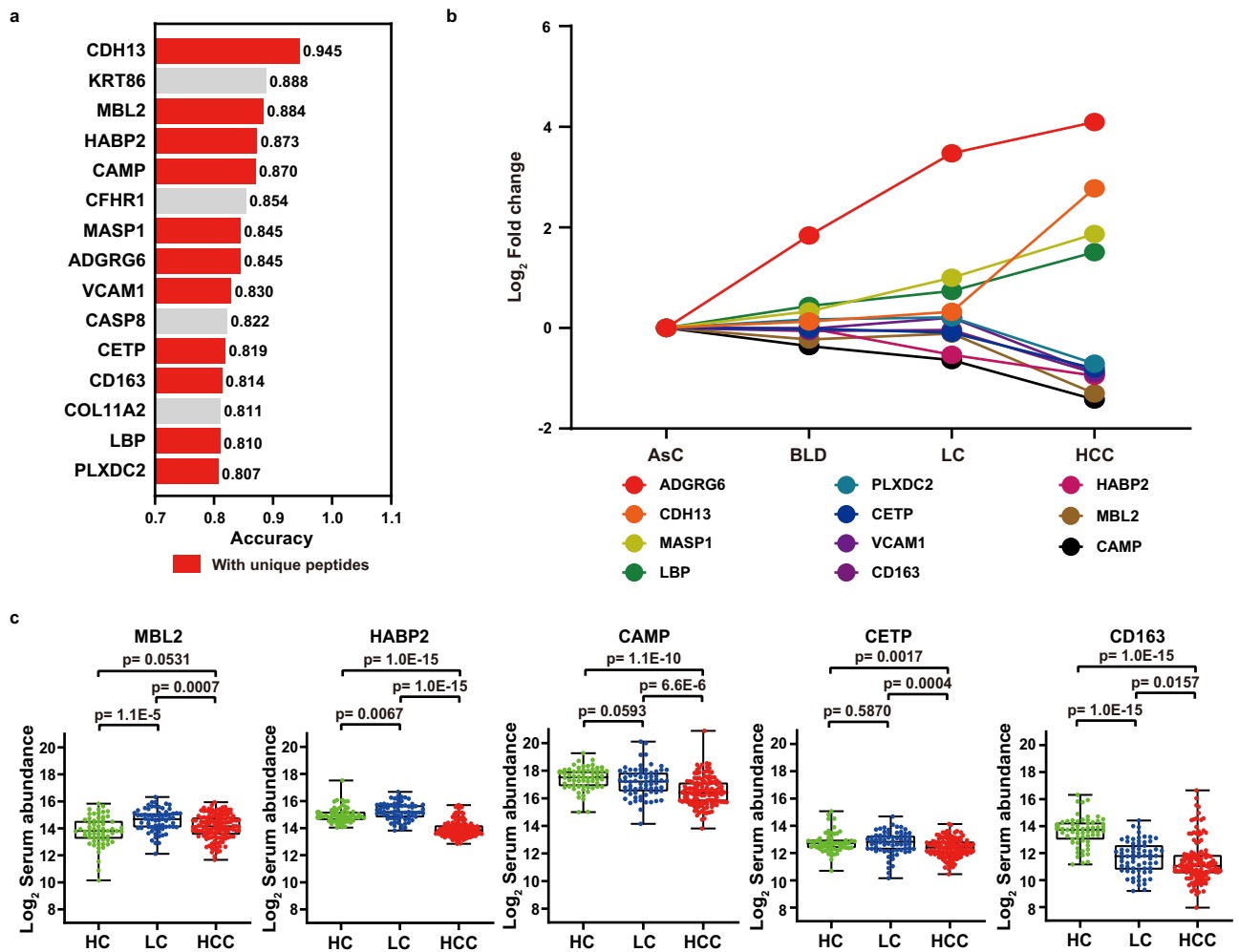

**Fig. 5 | Screening and validation of serum candidate biomarkers using PRM-targeted proteomics. a** Ranking of candidate biomarkers in the LVQ model with an accuracy of >0.8 in discriminating HCCs and LCs. The red color showed that the protein has unique peptides and could be used as PRM candidates. **b** Variation of fold changes of 11 candidate biomarkers for PRM target validation in four groups. **c** Comparison of PRM quantification of candidate biomarkers in HCC ($n = 130$), LC ($n = 68$) and HC ($n = 61$) groups. The quantitation data were Log$_2$ scaled. Significance was determined by two-sided Wilcoxon test with Benjamini-Hochberg multiple test adjustment. Box plots indicate median (middle line), 25%, 75% percentile (box), and minimum and maximum (whiskers) as well as outliers (single points). Source data are provided as a Source Data file.

combination in the diagnosis of HCC, AFP + PIVKA-II was used in comparison to P4 panel. The P4 panel had higher and more stable sensitivity than AFP + PIVKA-II in different clinical stages, especially in early HCC clinical stages like TNM I stage (0.875 vs 0.750), BCLC 0-A stage (0.902 vs 0.754), CNLC I stage (0.878 vs 0.683) (Fig. 6g, Supplementary Table 3). It suggested that P4 panel possessed a good predictive effect in early diagnosis of HCC patients.

**The P4 panel accurately predicted conversion of LC to HCC earlier**
As well known that imaging remains the gold standard for the diagnosis of HCC compared to the clinical protein biomarkers AFP, PIVKA-II, AFP + PIVKA-II, and other reported score-based models. To assess the efficacy of the P4 panel in detecting early-stage HCC and to compare it with other commonly used methods, we recruited 253 LC patients in a prospective clinical cohort to collect imaging data, PRM quantitative results of HABP2 + CD163, traditional protein biomarker assessment results (AFP and PIVKA-II) and the widely accepted ASAP risk score model (including age, sex, AFP and PIVKA-II) and aMAP risk score (including age, male, albumin, bilirubin and platelet) data at a series of follow-up time points with LC patients developing HCC as the final end-point. 253 LC patients of the modeling set were divided into a

training set ($n = 177$) and a testing set ($n = 76$) at a ratio of 7:3 randomly by random sampling (Supplementary Data 6). As expected, the P4 panel was effective in the diagnosis of the early stage HCC with highest AUC (0.890), sensitivity (0.909), and specificity (0.877), outperforming AFP, PIVKA-II, AFP + PIVKA-II, ASAP model and aMAP score (Fig. 7a, Supplementary Fig. 6a, Supplementary Table 4). The scores of P4 panel were significantly higher for LC patients who subsequently developed HCC than that for LC patients who did not develop HCC ($p < 0.0001$) (Fig. 7b). Significantly, in the validation cohort, more than 90% LC patients (90.9%, 10/11) who subsequently developed HCC were detected accurately, which was significantly higher than AFP (45.5%, 5/11), PIVKA-II (18.2%, 2/11), AFP + PIVKA-II (81.8%, 9/11), ASAP model (54.4%, 6/11) and aMAP score (72.7%, 8/11), suggesting that the P4 panel could be a good predictor for LC patients at the risk of developing HCC (Fig. 7c, Supplementary Fig. 6b-m). While there were 12.3% (8/65) of LC patients inconsistent with the prediction for conversion to HCC, which had high scores but none had HCC at the end of follow-up, suggesting that the P4 panel also suffered from the inevitable false positive rate of other strategies (12.3%).

Notably, the P4 panel detected more than 90% LC patients who subsequently developed HCC (90.9%, 10/11), completely consistent with the imaging diagnosis and is even earlier than the imaging ranging

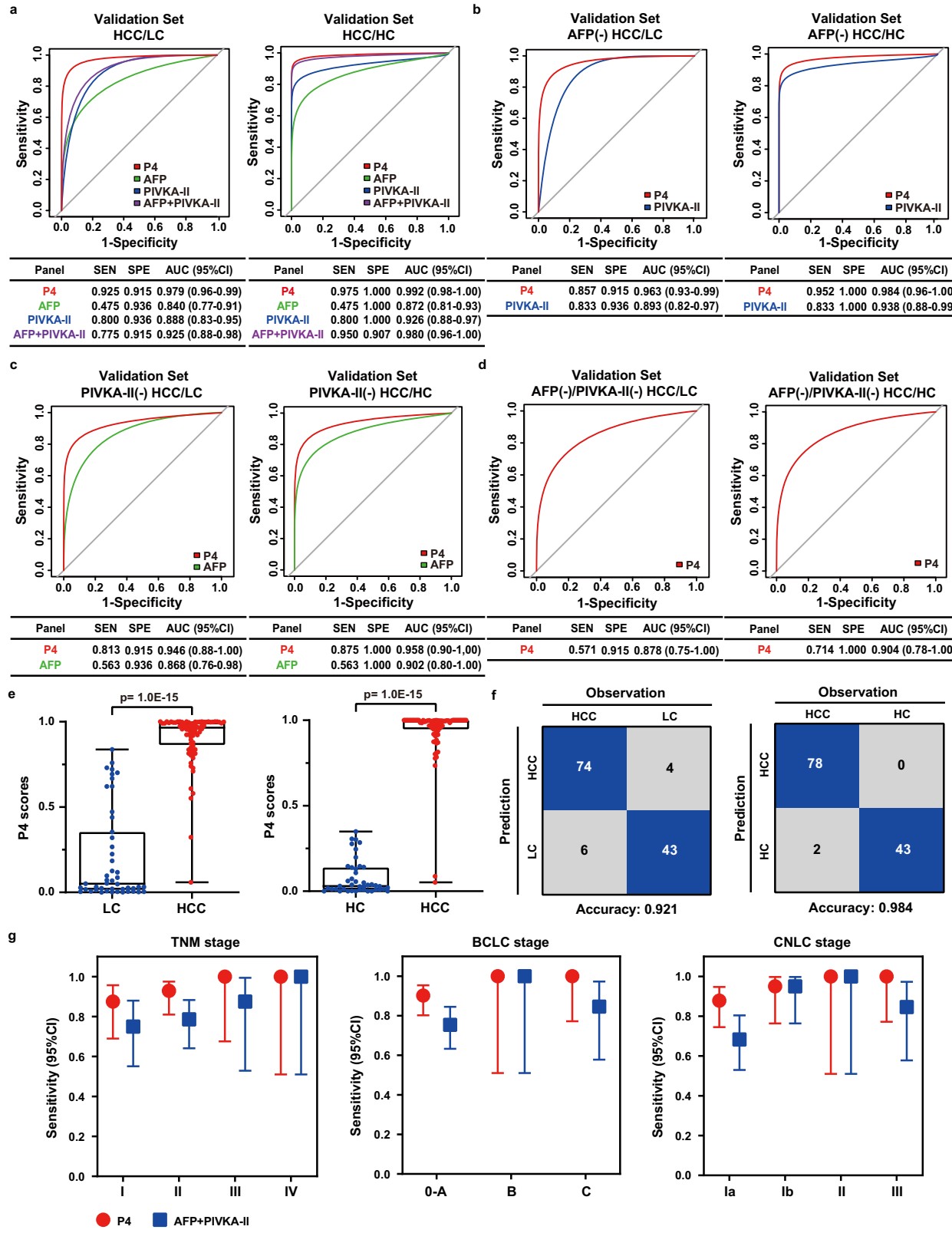

from 3.6 to 20.1 months, with a median of 11.4 months (Fig. 7d, e). The only patient whose HCC was not diagnosed earlier by the P4 panel was also not diagnosed using any other clinical biomarkers or other early diagnosis models. Furthermore, P4 panel had better concordance with positive imaging findings during follow-up (100% vs. 63.6%, 63.6%, 90.9%, 63.6%) compared to AFP, PIVKA-II, ASAP model, and aMAP scores at the follow-up periods (Fig. 7f, g). These findings suggest that

P4 panel could indeed be a promising predictor of conversion to HCC in LC patients compared to traditional protein biomarkers or other score-based models.

## Discussion

In current studies, many biomarkers based on liquid biopsy have been developed for the cancer diagnosis or monitoring, for example

**Fig. 6 | Diagnosis performance of the P4 model in validation cohort. a** ROC curves of P4 panel, AFP, PIVKA-II, and their combination for HCC patients ($n = 80$) versus LC patients ($n = 47$) and HCC patients ($n = 80$) versus HC ($n = 43$) in validation cohort. **b** ROC curves of P4 panel and PIVKA-II for AFP-negative HCC patients ($n = 42$) versus LC patients ($n = 47$) and HCC patients ($n = 42$) versus HC ($n = 43$) in validation cohort. **c** ROC curves of P4 panel and AFP for PIVKA-II -negative HCC patients ($n = 16$) versus LC patients ($n = 47$) and HCC patients ($n = 16$) versus HC ($n = 43$) in validation cohort. **d** ROC curves of P4 panel for AFP-negative and PIVKA-II-negative HCC patients ($n = 7$) versus LC patients ($n = 47$) and HCC patients ($n = 7$) versus HC ($n = 43$) in validation cohort. **e** Differences of P4 scores between HCC patients ($n = 80$) and LC patients ($n = 47$), and HCC patients ($n = 80$) and HC

($n = 43$). Significance was determined by two-sided Wilcoxon test with Benjamini-Hochberg multiple test adjustment. Box plots indicate median (middle line), 25%, 75% percentile (box) and minimum and maximum (whiskers) as well as outliers (single points). **f** Confusion matrix showed P4 panel performance for classifying HCC and LC, HCC and HC in the validation set. **g** Sensitivity with 95% confidence interval (CI) of P4 score and AFP + PIVKA-II in HCC of different clinical stages, such as TNM stages (Stage I, $n = 24$; Stage II, $n = 42$; Stage III, $n = 8$; Stage IV, $n = 4$), BCLC stages (Stage 0–A, $n = 61$; Stage B, $n = 4$; Stage C, $n = 13$) and CNLC stage (Stage Ia, $n = 41$; Stage Ib, $n = 20$; Stage II, $n = 4$; Stage III, $n = 13$). Error bars were defined to 95% CI of sensitivity. Source data are provided as a Source Data file.

methylated DNA[28–30], ctDNA[31,32] and microRNA[33,34]. However, nucleic acid biomarkers are economic- and time-constrained for clinical applications due to their complex detection methods and high requirements for sample preservation and handling[35]. Proteins, on the other hand, have good stability and can be easily developed into biomarkers for clinical applications[36]. While, MS-based screening of serum protein biomarkers has unique challenges due to the interference of high abundance proteins[37]. In this study, we replaced traditional DDA-MS with DIA-MS, which eliminated the need for cumbersome and expensive abundance protein reseparation and fraction in individual serum samples[38,39]. Although the depth of our serum proteome could be improved, we detected hundreds of proteins that are not available in the human plasma proteome database, e.g., serum amyloid A-1 protein and Proline-rich acidic protein 1. The high dynamic range of protein abundance in serum limits the sensitivity of MS-based proteomics, while a median CV of 18% in our assay is much better than the biological variation. Furthermore, the use of PRM-MS targeted proteomics validation method improves the accuracy of high-throughput validation, which can be used for antibody-free and batched validation of candidate biomarkers cost-effectively, and could be further optimized for clinical translation[40,41]. Therefore, our workflow is well suited to study tumor-related serum protein variations at a proteomic scale and provides an important resource for screening early diagnostic biomarkers for HCC.

In this study, we developed and validated a 4-serum-protein based panel for early diagnosis of HCC with sensitivity 0.925, specificity 0.915, and AUC 0.979. In an independent validation cohort, the panel was able to identify occult HCC that did not observed by imaging with more than 90% accuracy, although there was a certain rate of false positives (12.3%). The identification of high-risk populations for HCC by this panel that were diagnosed 1 year later through standard diagnostic methods demonstrated the utility of this panel for HCC screening and the potential for detecting high-risk populations for HCC through such screening. Therefore, we proposed that facile and scalable analyses of serum proteins based on serum proteomics could be used to prescreen high-risk populations for HCC to increase the accessibility of HCC detection and reduce unnecessary follow-up imaging procedures and invasive biopsies. In the current field of HCC screening, our initial idea on how to integrate the panel with the clinic in the future comes from three sources: (1) the panel can be further used as a complementary testing for patients with AFP-negative or PIVKA-II-negative but with high-risk clinical factors; (2) the panel can be used as a basis for further screening in patients who refuse imaging screening; (3) the panel can improve the detection rate of early HCC and serve as an alternative screening method for people at high risk of HCC.

Of course, we should acknowledge several limitations of this study. Firstly, multi-center and large-scale prospective clinical cohorts still need to be used to verify the universality of our model, including the sensitivity, specificity and accuracy of the model. Secondly, more healthy individual samples were needed to confirm the specificity of the assay in future studies. Thirdly, whether the P4 panel in clinical routine to enable early diagnosis of HCC is need further test. Finally, in

this study, we did not directly assess the model in patients with other cancer types, which can determine whether the model is specific to HCC.

In summary, our study presented an effective spectrometry (MS)-based proteomics workflow for the discovery and validation of early diagnosis serum biomarkers of HCC. We developed an earlier and more accurate predictive panel for the conversion of LC to HCC than existing clinical methods, which may provide useful reference for the early diagnosis of HCC.

## Methods

### Patient cohorts and sample collection

This project was approved by the Institution Review Board of Mengchao Hepatobiliary Hospital of Fujian Medical University. Informed consent was obtained from each participant before the operation. The use of clinical specimens was completely in compliance with the "Declaration of Helsinki".

For constructing the discovery cohort, 40 chronic asymptomatic hepatitis B virus carrier (AsC), 64 basic liver diseases (BLD) (including 16 chronic hepatitis B (CHB), 18 ALD, and 30 NAFLD samples), 53 liver cirrhosis (LC) and 163 hepatocellular carcinoma (HCC) serum samples of patients were enrolled from April 2014 to June 2021 in local hospital. The inclusion criteria for HCC patients were: 1) Histopathologically confirmed hepatocellular carcinoma; (2) Radical hepatectomy performed[2]; (3) No preoperative anti-cancer treatment; (4) had complete physio-biochemical clinicopathological data. Patients with CHB, ALD, NAFLD, and LC required physiological and imaging evidence for the diagnosis. Gender, detailed physiological and biochemical indicators for all patients, as well as clinicopathological characteristics for HCC patients can be found in the Supplementary Data 1.

The retrospective validation cohort collected serum samples from 210 HCC patients, 115 LC patients and 104 healthy individual controls from June 2016 to July 2021 in local hospital, and candidate biomarkers were validated using PRM-MS. Detailed inclusion and exclusion criteria of the patients were the same as for the discovery cohort.

Notably, 253 LC patients were prospectively collected from the clinical trail of Early Screening for Hepatocellular Carcinoma program ($n = 17$, Ethical Approval No.: Keshen 2018_005_02) and the PreCar program ($n = 236$, Prospective suRveillance for very Early hepatoCellular cARcinoma project, ClinicalTrials.gov No.: NCT03588442) from June 2018 to July 2022, in which the enrolled LC patients were followed up every 6 months with AFP, liver function tests, a chest computed tomography (CT) and ultrasound or a contrast-enhanced CT scan or magnetic resonance imaging (MRI) of the abdomen at each visit. The diagnosis of HCC followed the strict criteria of the European Association for the Study of the Liver (EASL). The median follow-up time of LC patients was 43.9 months and 36 LC patients were diagnosed with HCC during subsequent follow-up. The median time from enrollment to progression to HCC for these LC patients was 28.1 months.

All samples were collected from clinical specimen banks of Mengchao Hepatobiliary Hospital of Fujian Medical University. Serum samples were collected using an intravenous tube without

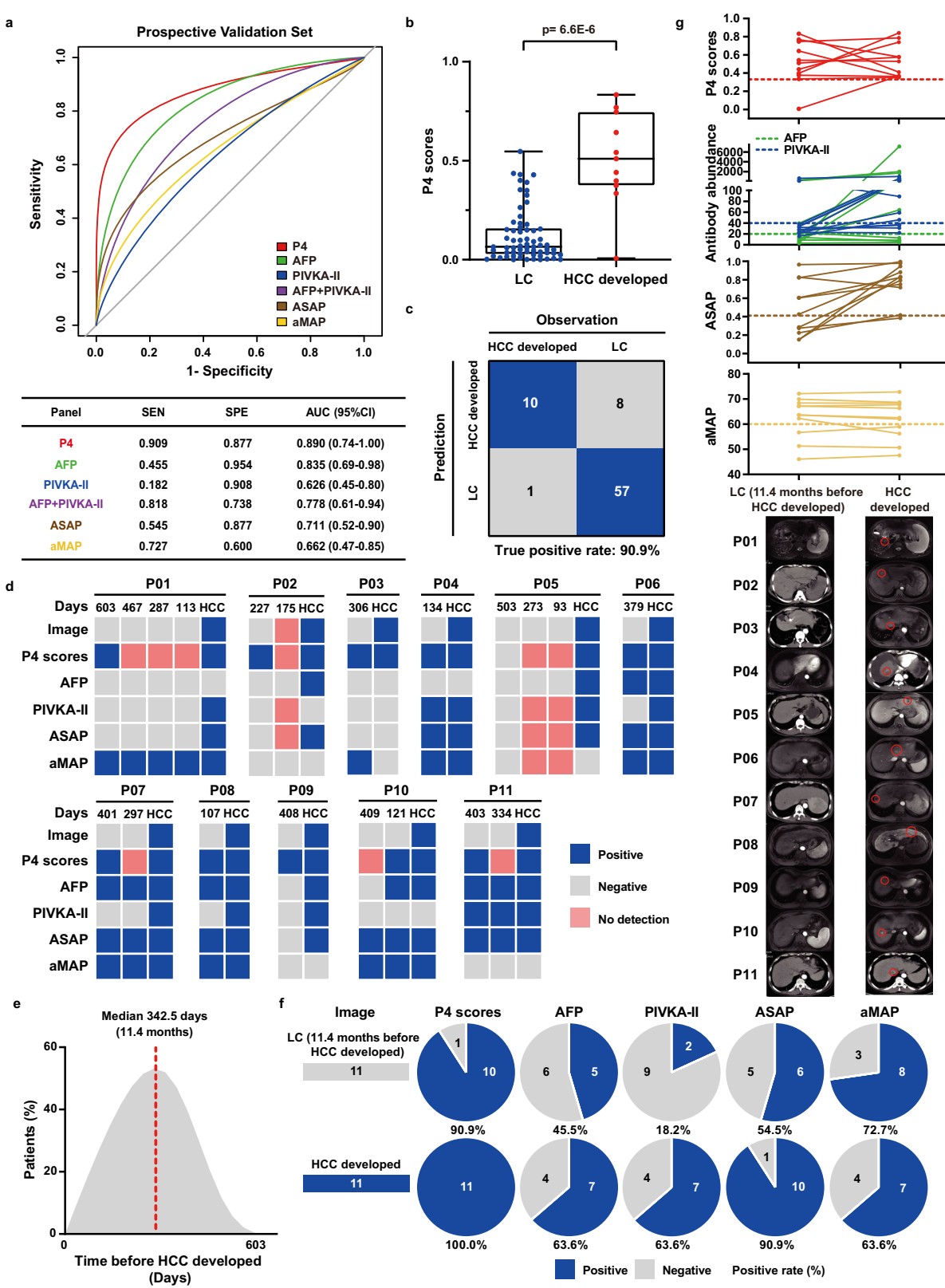

anticoagulant and coagulated at room temperature for 30 min, then centrifuged at 1000 g for 10 min. The supernatant serum was collected and frozen at −80 °C for subsequent use.

## Separation of LAPs and HAPs for serum samples

To construct a spectrum library for DIA-MS, 40 samples were randomly selected and mixed equally into one pool, so 320 samples were divided into 8 pools in total. HAPs and LAPs in serum were separated by high performance liquid chromatography (HPLC) using Human 14 Multiple Affinity Removal System Column (Agilent Technologies, Santa Clara, CA, USA) according to the manufacturer's instructions. The collected fractions of LAPs and HAPs were concentrated into one tube by 3 K cutoff centrifugal filter for protein concentration measurement using the BCA assay, respectively[42].

**Fig. 7 | Performance of the P4 model in predicting people at high risk of HCC in prospective validation cohort. a** Performance of the P4 score, serum biomarkers (AFP, PIVKA-II, AFP + PIVKA-II), and early diagnosis score models (ASAP and aMAP score model) for LC patients ($n = 76$) in prospective validation set to predict LC patients who developed to HCC at subsequent follow up. The upper panel illustrated ROC curves, and the lower panel showed the AUC, sensitivity, and specificity. **b** Differences of P4 scores between LC patients who developed HCC ($n = 11$) and LC patients who did not develop HCC ($n = 65$) in the validation cohort. Significance was determined by Wilcoxon test with Benjamini-Hochberg multiple test adjustment. Box plots indicate median (middle line), 25%, 75% percentile (box) and minimum and maximum (whiskers) as well as outliers (single points). **c** Confusion matrix showed P4 panel performance for predicting people at high risk of HCC in the validation cohort ($n = 76$). **d** The categorization of imaging results, P4 scores, serum biomarkers, ASAP model and aMAP score results of 11 LC patients in the validation cohort who developed HCC during follow-up was shown in each color-code plot. Blue indicated positive, gray indicated negative, while pink indicated no detection. **e** Time distribution of P4 panel predicted HCC occurrence earlier than imaging results. **f** The concordance comparison of P4 scores, serum biomarkers and risk scores compared with positive and negative of CT/MRI scan results during HCC occurrence. **g** The time-course demonstration of imaging results and quantified levels of P4 scores, serum biomarkers and risk scores during the clinical course of 11 patients who developed HCC. The corresponding cutoffs were indicated by dashed lines. Source data are provided as a Source Data file.

### Protein digestion

Protein digestion for serum samples were performed using a modified Filter-Aided Sample Preparation (FASP) method[43]. To prepare the LAPs and HAPs peptides for MS detection, 500 μg protein samples were diluted by 400 μL lysis buffer (6 M urea and 1× protease inhibitor cocktail). Then, 8 mM dithiothreitol was added to reduce for 30 min at 55 °C, followed by 50 mM iodoacetamide (IAA) to alkylate for 30 min in dark at room temperature. The protein solutions were then washed twice with 100 mM tetraethyl-ammonium bromide in 3 K cutoff centrifugal, and digested using trypsin at a concentration of 1:50 (w/w) for 18 h at 37 °C. Digested peptides are centrifuged at 14,000 g for 30 min and collected in the outer tube of the filter. Finally, the digested peptides were eluted and evaporated to dryness for LC-MS/MS analysis.

For peptide preparation of DIA-MS and PRM-MS, 10 μL of each individual serum sample was diluted by adding 200 μL lysis buffer, and 200 μg proteins were prepared in the same method as mentioned above for subsequent MS analysis, respectively.

### High pH reversed-phase separation

LAPs and HAPs were fractioned by an offline LC system (Acquity UPLC, Waters, the U.S.A) via high pH (pH = 10) separation, which was performed in $C_{18}$ reverse phase column (2.1 mm × 50 mm, 1.7 μm, catalog No.186002350, Waters, the U.S.A) with a flow rate of 400 μL/min. Peptide mixtures were resuspended in mobile phase A and eluted with a subsequent and linear gradient as following: 0–5 min, 100% mobile phase A; 5–20 min, 93% mobile phase A; 20–25 min, 65% mobile phase A. Mobile phase A was $H_2O$ with 0.1% FA and mobile phase B was ACN with 0.1% FA. Starting at 1 min, one tube of fractions was collected every 30 s. Thus, 48 tubes of LAPs or HAPs fractions were collected for each of the 25 min gradients. To reduce the MS detection time, we mixed these fractions with the same time interval. For example, fractions collected in 1, 4, 7, 10, 13, and 16 min were mixed into one tube. Finally, a final of eight samples were dried by vacuum centrifugation for proteomic analysis.

### Data acquisition by DDA mass spectrometry

For spectrum library construction, 1 μg of each fraction was added to the nano liquid chromatography (Easy-nLC 1000, Thermo, the U.S.A.) which was linked with a Quadrupole-Orbitrap mass spectrometer (Q Exactive plus, Thermo, the U.S.A.). Briefly, the peptide was resuspended in mobile phase C (0.1% FA in water) and equal amounts of indexed retention time (iRT) peptide standards (Biognosys, Switzerland) were spiked into each sample. And the peptides were separated onto the $C_{18}$ analytical column (75 μm × 250 μm, 1.8 μm, catalog No. 186008818, Waters, the U.S.A) with a 70 min gradient at a constant flow rate of 300 nL/min (0–3 min, 4–7% of mobile phase D; 3-45 min, 7–14% mobile phase D; 45–60 min, 14–30% of mobile phase D, 60–70 min, 30–90% of mobile phase D and held at 90% mobile phase D for 15 min. Mobile phase C was 0.1% FA in water, mobile phase D was 0.1% FA in ACN). Mass spectrometry was operated under a DDA mode with 1.9 kV electrospray voltage at the inlet. DDA scheme included a full MS scan from 300 to 1,800 m/z at a 70,000 resolution (at m/z of 200) using an AGC target value of 3E6. Fifteen most intensive precursors of MS/MS scan were selected for high energy collisional dissociation (HCD) with 27% normalized collision energy. MS/MS spectra were acquired at resolution of 17,500 (at m/z of 200) using an AGC target value of 1E5 and a maximum injection time (IT) of 45 ms. Dynamic exclusion was applied with a repeat count of 1 and an exclusion time of 30 s.

### Data acquisition by DIA mass spectrometry

For data acquisition of DIA-MS, equal amount of iRT peptide standards were spiked into individual sample peptide, and nano-LC MS/MS basic parameter settings were the same as that for DDA. We used the experimental setting of the ordered allocation with HCC samples processed first, followed by LC, CHB, and AsC; while, within-group samples were analyzed in a complete randomized allocation. In DIA mode, DIA scheme was included a full MS scan from 400 to 1200 m/z at a 70,000 (at m/z of 200) resolution and 32 MS/MS scans were acquired with a 17,500 resolution at a m/z of 200 and a max IT of 20 ms. The cycle of 32 MS/MS scans (center of isolation window) with two kinds of wide isolation window are as follows (m/z): 410–990 m/z with 20 m/z wide and 1050–1150 m/z with 100 m/z wide. The dynamic exclusion time was set to 20 s.

### Construction of spectral library and analysis of DIA-MS

For spectrum library construction, both DDA and DIA files were processed using Spectronaut (Version. 13.2.19)[44]. The background database was built with FASTA file of *Homo sapiens* containing 20368 reviewed proteins (Published by Uniprot in 2020 year), combining with the fusion sequence of iRT. Digest enzyme was trypsin/P and max missed cleavages only allowed 2. Carbamidomethyl was set to fixed modification, and acetyl (Protein N-term) and oxidation were set to variable modification. The false discovery rate was set to 1% at peptide precursor level and 1% at protein level. For the quantitative analysis of proteins across the 320 serum samples, 320 DIA raw data files were searched against the hybridized spectral library followed by the quantification via Spectronaut Pulsar X. Q value cutoff of protein and precursor were both set to 0.01.

### Quality control of the mass spectrometry platform and the serum proteomics experiment

To evaluate the performance of the mass spectrometry systems, the Hela standard peptides (Pierce, the U.S.A) was measured in the process of the project as the quality-control standard. DDA analysis was interspersed per two experimental samples and DIA-MS analysis was interspersed per ten experimental samples. The standard was analyzed using the same method and conditions as using in the HCC-related serum samples. A Pearson's correlation coefficient was calculated for all quality-control runs based on package reshape2 (Version.1.4.4) and corrplot (Version.0.92) of R (Version.4.0.2).

To evaluate the quantitative accuracy of the serum proteome experiments, technical replicates of six serum samples (HCC, $n = 2$; LC, $n = 2$; CHB, $n = 2$) were also measured in the middle and at the end of

the project by DIA-MS. The reproducibility of two replicates with the same samples was assessed by correlation analysis and CV calculation. A Pearson's correlation coefficient was calculated for all replicated runs based on package reshape2 and corrplot of R. CV calculation was based on the ratio of the standard deviation (SD) to the mean of the protein quantification, and the visualization of the violin plot was performed by Graphpad Prism (Version.8).

## Data processing for serum proteomics data
The pre-processing of the proteomic data was performed using wkomics (https://www.omicsolution.com/wkomics/main/) analysis platform[45]. For proteins with ≥ 40% integrity in HCC or non-HCC samples, missing values were filled with the SeqKNN method; while when integrity <40% in both two groups, missing values were filled with a minimal value. Then median normalization and $Log_2$ transformation were performed for subsequent data analysis.

## Identification and functional analysis of HCC-related differentially abundant proteins
Differentially abundance proteins were identified using two-sided independent sample t-test and Benjamini-Hochberg multiple test adjustment based on wkomics analysis platform. Proteins with $p < 0.05$ and FC ≥ 1.2 were eligible as differentially abundant proteins. Gene ontology (GO) analysis and Kyoto Encyclopedia of Genes and Genomes analysis were used to enrich the functions and pathways of HCC-related differentially abundant proteins. The statistical difference of the enrichment was evaluated by the hypergeometric test and the method of Benjamini-Hochberg multiple test adjustment. Protein-protein interaction (PPI) network analysis was performed using online analysis tool String (https://cn.string-db.org/). Proteins clustering in PPI was based on K-means algorithm and visualization of the network diagram used mapping tool Cytoscape 3.9.1.

## Selection of unique peptides of HCC candidate diagnosis biomarkers
Learning Vector Quantization (LVQ) model[46] achieved via package mlbench and caret of R was used to evaluate the accuracy of single protein in identifying HCC patients for HCC-related differentially abundant proteins. The discovery cohort of 320 patients was randomly divided into ten equal data sets and repeated ten times to train the LVQ model. And the candidate proteins were selected with accuracy higher than 0.8. The ionic characteristics of peptides matched candidate proteins were continued to evaluate as follows: the selected peptides were required to be unique and without modification; the best length of peptides was 8–12 amino acids and have at least 5 ions with the intensity more than $10^4$. Each protein ultimately selected one unique peptide as a candidate for PRM-MS validation.

## Validation of candidate biomarkers using targeted proteomics
PRM quantification strategy was used to further validate the candidate biomarkers identified by DIA-MS above. The synthesized isotope labeled peptides of the candidate unique peptides were spiked into peptide samples for absolute quantification. The mixed peptides were loaded into LC-MS/MS for data acquisition in PRM mode at a flow rate of 300 nL/min. The LC gradient started with 92% of phase C and decreased to 82% at 46 min. Phase C then reached 68% at 51 min. The gradient finally reached 20% C at 52 min and was held for 5 min until next injection. A full MS survey scan was set from 300 to 1800 m/z at a resolution of 70,000 (at 200 m/z) with 1.9 kV electrospray voltage at the inlet. Target ions were submitted to MS/MS by HCD with MS/MS spectra resolution of 17,500 (at 200 m/z) and 1 m/z isolation window. PRM transitions were extracted from PRM-MS raw data files and analyzed using Skyline (Version.3.6.0.1)[47]. Peptide peak areas were calculated as the sum of at least three most abundant transitions. Lists of all peptides targeted in the PRM analyses are provided in Supplementary Table 2.

## Random forest model construction and performance evaluation for HCC early diagnosis biomarkers
Machine learning model based on Random forest algorithm[48] was constructed by package randomForest (Version.4.6-14) of R to predict diagnostic performance for different biomarker panels. 259 samples (containing 130 HCC, 68 LC, and 61 HC) of the modeling set were divided into a training set ($n = 182$, containing 93 HCC, 47 LC, and 42 HC) and a testing set ($n = 77$, containing 37 HCC, 21 LC, and 19 HC) at a ratio of 7:3 randomly by random sampling. In the training of the model, the number of decision trees (ntree) was set to 500 and its number of variables (mtry) was set to half of the number of classifiers according to different panels (non-integer values were rounded). The trained models were then tested internally in the testing set to obtain corrected thresholds for distinguishing HCC patients from LC patients and HC individuals. The stability validation of models was performed in an extra validation set consisting of 80 HCC, 47 LC, and 43 HC. The performance of different models was assessed by comparing sensitivity, specificity, and area under the curve (AUC) of receiver operating characteristic (ROC) curve. ROC curves were achieved by package pROC (Version.1.18.0) of R.

## Random forest model construction and performance evaluation for HCC early diagnosis panels in the prospective validation cohort
Random forest model for HCC early diagnosis was constructed by the same method as above. A prospective validation cohort containing 36 LC patients with HCC development and 217 LC without HCC development was divided into a training set ($n = 177$) and a validation set ($n = 76$) at a ratio of 7:3 randomly by random sampling. The training set was used to model training for HCC early diagnosis biomarker panel. The trained models were then verified in the validation set to assess its prediction performance ability for LC patients with HCC development. The performance of different models was assessed by comparing sensitivity, specificity, and AUC of ROC curve. ROC curves were achieved by package pROC of R. In addition, two clinical HCC risk prediction models, ASAP (including age, sex, AFP, and PIVKA-II)[49] and aMAP (including age, male, albumin, bilirubin, and platelet)[50], were used to compare with the P4 panel in LC patients with HCC development.

## Statistics and reproducibility
No statistical method was used to determine sample sizes. Experiments were not randomized. As samples were required for both classifier training and validation, they were randomly allocated into two subsets at a ratio of 7:3. Data distributions were assumed to be normal, but this was not formally tested. No data were excluded from the analyses. AUC was used to measure the performance of biomarkers in distinguishing HCC and LC patients. All statistical tests were two-sided, and unless stated otherwise, the results were considered as significant at a $p$ value threshold of 0.05. Details of statistical analyses were provided throughout the text and in figure legends with their associated sample sizes. Graphpad Prism version 8 was used for drawing scatter plot, box plot, violin plot, and column for data visualization. Further information on research design is available in the Nature Research Reporting Summary linked to this article.

## Reporting summary
Further information on research design is available in the Nature Portfolio Reporting Summary linked to this article.

## Data availability

The raw data of mass spectrometry generated in this study have been deposited to the ProteomeXchange Consortium (http://proteomecentral.proteomexchange.org) via the iProX partner repository (https://www.iprox.cn/, Project ID: IPX0005766000)[51,52] with the dataset identifier PXD046887. The iProX data is publicly available (www.iprox.org, accession number IPX0005766000). All relevant data are included in the manuscript and the Supplementary Information. Source data are provided with this paper.

## Code availability

Data analysis was performed in R version 4.0.2. using custom or publicly-available R package. Individual packages are explicitly cited in the manuscript. The code is available upon request and deposited in a Github repository. We have obtained a (https://doi.org/10.5281/zenodo.10117967) for the Github repository at Zenodo[53].

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

## Acknowledgements

This work was supported by the Scientific Foundation of Fujian Provincial Health Commission (Grant No. 2021ZQNZD014), the Major Science and Technology Special Project of Fujian Province (Grant No. 2022YZ036012); the Major Projects of Medicine and Health in Zhejiang Province (WKJ-ZJ-2306), the Natural Science Foundation of Fujian Province (Grant No. 2023J011478); the Scientific research project of Health and Family Planning Commission of Fujian province (Grant No. 2022GGA051), and the Scientific Foundation of Fuzhou Health Department (Grant No. 2022-S-009).

## Author contributions

X.L. and X.X. led this project in generating proteomics data, data analysis, data validation and manuscript preparation. Y.W. and Y.Z. performed the collection and provision of clinical samples. L.C., E.H., Z.L., and C.H. contributed to the collection and collation of clinical data. L.C., J.O., F.W., Z.L., and M.L. contributed to proteomics sample preparation of serum. X.X., L.C., and J.O. coordinated mass spectrometry data acquisition. X.X. and L.C. performed proteomics data analysis and statistical analysis. L.C. and J.O. constructed the data portal; L.W., J. L., and X.L. supervised the project, revised and reviewed the manuscript. All the authors contributed to the manuscript revision.

## Competing interests

The authors declare no competing interests.
