## [Peer Review File · Nature Communications]

This manuscript has been previously reviewed at another journal that is not operating a transparent peer review scheme. This document only contains reviewer comments and rebuttal letters for versions considered at *Nature Communications*.

REVIEWERS' COMMENTS

Reviewer #2 (Remarks to the Author):

I have no additional comments.

Reviewer #3 (Remarks to the Author):

The authors have done a nice job responding to my comments from my previous review. The inclusion of healthy controls is very nice.

I have only two minor comments:

- 1) I still cannot find the raw data. The link provided in the rebuttal (<https://www.iprox.cn/page/SSV024.html?url=1676252260503XUB6>) goes to an empty webpage. If I search for the id (IPX0005766001), I receive the message "No data available in table".
- 2) I do not see any info in the Methods about the "ordered allocation" sample run order that is described in the rebuttal. The authors should include this in the manuscript for transparency.

RESPONSE TO REVIEWERS' COMMENTS

We sincerely appreciate the efforts from the reviewers to improve our manuscript. The arisen issues by two Reviewers have been addressed point by point as follows:

Reviewer #2 comments:

I have no additional comments.

Response: Thanks very much.

Reviewer #3 comments:

The authors have done a nice job responding to my comments from my previous review. The inclusion of healthy controls is very nice.

Response: Thanks very much.

1) I still cannot find the raw data. The link provided in the rebuttal (<https://www.iprox.cn/page/SSV024.html?url=1676252260503XUB6>) goes to an empty webpage. If I search for the id (IPX0005766001), I receive the message "No data available in table".

Response: Thanks very much. We don't make the raw data public until the article is officially accepted, so you'll need to enter a password "juw9" when using the link (<https://www.iprox.cn/page/SSV024.html?url=1676252260503XUB6>). This password we have provided in rebuttal. Now, we make sure the raw data is released and publicly accessible and downloadable using this ID (IPX0005766001) or ProteomeXchange identifier PXD046887.

2) I do not see any info in the Methods about the "ordered allocation" sample run order that is described in the rebuttal. The authors should include this in the manuscript for transparency.

Response: Thanks very much. We have added this part to the method of the revised manuscript. (P15, Line 442-444)